# The Human Microbiome as a Therapeutic Target for Metabolic Diseases

**DOI:** 10.3390/nu16142322

**Published:** 2024-07-19

**Authors:** Thi Phuong Nam Bui

**Affiliations:** Department of Experimental Vascular Medicine, Amsterdam University Medical Center, 1105 AZ Amsterdam, The Netherlands; t.p.n.bui@amsterdamumc.nl

**Keywords:** human microbiota, host metabolism, microbial metabolites, short chain fatty acid, bile acid, metabolic disease

## Abstract

The human microbiome functions as a separate organ in a symbiotic relationship with the host. Disruption of this host–microbe symbiosis can lead to serious health problems. Modifications to the composition and function of the microbiome have been linked to changes in host metabolic outcomes. Industrial lifestyles with high consumption of processed foods, alcoholic beverages and antibiotic use have significantly altered the gut microbiome in unfavorable ways. Therefore, understanding the causal relationship between the human microbiome and host metabolism will provide important insights into how we can better intervene in metabolic health. In this review, I will discuss the potential use of the human microbiome as a therapeutic target to improve host metabolism.

## 1. Introduction

The human microbiome resides in our gastrointestinal tract and creates a dynamic and complex microbial ecosystem consisting of more than 1000 microbial species [1] and their phages. Based on epidemiological and omics studies combined with in vitro studies using various cell models and in vivo studies in mice, it appears that human health and disease risk may be mediated by the human microbiome [2]. In infancy, the composition and function of the human microbiome are influenced by the mode of birth and feeding, and it becomes stable by the age of three [3]. In adult life, these microbes are mainly influenced by lifestyle, medication and host genetics [4]. The gut microbiota, in turn, produce microbial components that act not only on local cells in the gut but also on peripheral tissues via the systemic circulation, playing a crucial role in training our immune system and regulating gut endocrine function and neurological signaling [2,5]. They are also involved in modifying drug action and metabolism, eliminating toxins and producing numerous signaling compounds. From time to time, we learn more about new functions of the gut microbiome.

There is an increasing global prevalence of metabolic diseases associated with unhealthy lifestyles [6]. These include type 2 diabetes (T2D), metabolic dysfunction-associated steatotic liver disease (MASLD), hypertension, hyperlipidemia and obesity. In 2019, the world was estimated to have around 44 million cases of T2D and 1.2 billion cases of MASLD [6], and these numbers are increasing rapidly. This rapid increase has been attributed to the overuse of processed foods, urbanization, smoking addiction and physical inactivity, resulting in a significant increase in people with poor metabolic health [7,8,9]. Despite the wide variation in the pathologies of these common metabolic disorders, they are all associated with abnormalities in the composition and function of the human microbiota [10,11,12]. It remains questionable whether there is a causal relationship between host metabolism and the microbiome. To date, results obtained from animal and fecal microbiota transplantation studies have demonstrated causal effects of the microbiome on host health [13,14,15]. Importantly, recent developments in next-generation microbiome sequencing to obtain comprehensive gene catalogues combined with targeted bioinformatics have provided a substantial amount of new knowledge on the role of the gut microbiota in human metabolism. Here, the implications of the human microbiome as a therapeutic target for metabolic disease will be discussed.

## 2. Interplay between the Human Microbiota, Diet and Health Determines Metabolic Outcome

The gut microbiome has been shown to play a critical role in gut homeostasis and beyond. Several cohort studies have reported profound associations between the human microbiome and host health [2,16,17]. These associations are strongly influenced by environmental factors, with diet being one of the most dominant drivers [18], creating a complex interplay between the human microbiome, diet and the host that determines the health outcomes (Figure 1). Diet provides nutrients to both the host and the microbiome through the digestion and ingestion of food components, and the microbial fermentation of indigestible food components, respectively. More importantly, there is considerable crosstalk between the host and the microbiome, with the host producing components for microbial activities such as mucin, whereas the microbiome uses both host-derived and dietary components not only for growth but also for the production of components that affect host health. Thus, diet shapes the microbiome composition and metabolic activities navigating towards either beneficial or detrimental effects [19], and these effects, in turn, are strongly mediated by the human microbiome [20].

Consuming a healthy diet, such as one rich in fiber, benefits our health by modulating the human microbiome to produce beneficial compounds such as short chain fatty acids [21]. Moreover, fiber has been shown to improve bowel function as well as intestinal function and transit time [22], thereby increasing microbial metabolism and mucosal turnover [23]. Similarly, low microbiome gene richness and clinical phenotypes have been increased by dietary intervention in obese and overweight individuals [18]. A recent human study showed that supplementation with fermented foods reduced markers of inflammation, and high-fiber dietary intake increased microbiota carbohydrate active enzymes [20]. Surprisingly, increased pro-inflammatory cytokines were observed in subjects with low baseline microbial diversity in the high-fiber diet group. This is an interesting observation as it is generally accepted that fiber intake improves host health via bacterial fermentation to produce short chain fatty acids, signaling molecules with important metabolic functions [21]. It may be necessary to increase the microbial diversity in the gut prior to fiber ingestion to achieve health benefits. Not only the type of food intake but also the eating pattern affect the microbiome. Intermittent fasting has been shown to modulate the gut microbiota and improve obesity and host energy metabolism [24].

In contrast to the dietary benefits of microbiome modulation, there are also adverse effects of unhealthy diets via gut microbial activities; for example, consumption of ultra-processed foods may increase the risk of cardiometabolic disease [25]. Diets that are largely heat-processed contain substantial amounts of advanced glycation end products, which are the result of a cross-linking non-enzymatic reaction between reducing sugars and free amino groups of proteins, nucleic acids or lipids. Dietary advanced glycation end products have been shown to contribute to increased inflammation, oxidative stress and modulation of insulin sensitivity in overweight individuals [26]. The excessive consumption of refined sugars has been correlated with adverse effects on the gut microbiome, including reduction in signaling molecules (short chain fatty acids), altered gut barrier function and increased inflammation, leading to various metabolic disorders [27]. In fact, it has been shown that ethanol derived from the human microbiome activities contributed to the development of MASLD [28]. In a human study, artificial sweeteners such as sorbitol were found to induce glucose intolerance by altering the gut microbiota [29]. It has become clear that the balance between the microbiome, the diet and the host is important for maintaining host health. Once this balance is disturbed, changes in both the microbiome and diet may be required to achieve optimal health benefits.

## 3. Altered Microbiota in Metabolic Syndrome

A healthy microbiome is characterized by a high diversity of microbial taxa, high microbial gene richness and stable microbial functional cores [17]. Several cohort studies have shown that the microbiome is altered in metabolic diseases as compared to healthy controls (Figure 2). It has been observed that the microbiome of obese individuals differs significantly from that of healthy individuals. It was first suggested that the gut microbiota of obese mice and obese humans had a higher ratio of members of the phylum Firmicutes to members of the phylum Bacteroides compared to that of lean counterparts [30]. In addition, obese individuals had a lower microbial diversity than that of lean individuals, but other studies in humans and rodents have found no difference in this ratio in obese versus lean individuals, and weight loss had no effect on this ratio [31,32,33]. Therefore, the Firmicutes/Bacteroides ratio may not be a good marker of obesity. At the species level, some species were associated with a high BMI, such as *Eubacterium ventriosum* and *Roseburia intestinalis* [34], while other butyrate producers and the methanogenic archaeon *Methanobrevibacter smithii* may be associated with leanness [35]. Another metagenome-wide association showed that the abundance of *Bacteroides thetaiotaomicron* was significantly reduced in obese compared to lean individuals, and the administration of this bacterium protected against obesity in mice [36]. In two independent cohorts, structural variations carrying *myo*-inositol metabolic genes in the *Anaerostipes hadrus* genome showed inverse correlations with body weight and body mass index [37]. It has been shown that obesity can be transmitted from humans to mice via fecal microbiome transplantation in a diet-dependent manner [38], suggesting opportunities to intervene in whole-body energy metabolism by targeting the microbiome.

Like obesity, T2D is one of the most common endocrine disorders with increasing prevalence and incidence, affecting nearly 15% of the adult population [39]. The etiology of T2D involves multiple genetic and environmental factors, including the human microbiome. The association between the human microbiome and type 2 diabetes was first reported in 2012 [10]. Following this study, the richness of the human microbiome was found to correlate with several metabolic biomarkers [40]. Many cross-sectional studies have shown that the microbiome of T2D subjects has reduced microbial gene diversity, with a decrease in butyrate-producing species and *Akkermansia muciniphila*, and an increase in bacterial strains with pro-inflammatory functional potential [10,41]. To date, a few bacterial strains have been found to improve insulin sensitivity and glycemic variability in metabolic syndrome, obesity and T2D [13,42,43]. Metformin, a commonly used drug for diabetes, has been shown to favorably alter the gut microbiome, contributing to its therapeutic effects [44]. Potential bacterial candidates that influence metformin therapy have been identified using a host–microbe–drug–nutrient screen [45]. Statin therapy has been found to be associated with a reduced prevalence of gut microbiota dysbiosis [46]. Future research is needed to determine whether there are synergistic effects of conventional drugs and microbiome-based therapeutics or multiple strains inT2D.

Recent research has suggested a role for the human microbiome in the pathogenesis of MASLD [47], and there is growing evidence of a perturbed microbiome in MASLD patients, with increased abundance of *Anaerobacter*, *Streptococcus*, *Escherichia* and *Lactobacillus* and decreased abundance of *Prevotella* and *Alistipes* as compared to healthy controls [48]. It has been shown that the microbiota of MASLD patients had an enrichment of *Lactobacillus*, which is responsible for endogenous ethanol production and contributes to the pathogenesis of the disease [28]. Indeed, it has been described that ethanol activates nuclear factor-kB (NF-kB) signaling pathways and impairs intestinal barrier function [49], potentially causing oxidative damage to hepatocytes, which may induce hepatic inflammation. Supplementation with an ethanol-producing strain accelerated MASLD pathogenesis in mice [50]. It remains unknown whether there are gut bacteria that can consume ethanol and whether supplementation with these ethanol-consuming strains can reduce endogenous ethanol production and thereby prevent the progression of MASLD.

## 4. The Human Microbiome as Therapeutic Target to Improve Metabolic Health

Accumulating evidence from both humans and rodents supports the hypothesis that low-grade, systemic and chronic inflammation induced by the gut microbiota may (partially) drive metabolic diseases such as obesity, diabetes and MASLD in humans. This provides the rationale for the ongoing intensive search for gut microbial messengers and gut bacteria that regulate whole-body energy metabolism. Impaired gut barrier function has been implicated in the pathogenesis of several diseases including type 1 diabetes and inflammatory bowel disease (IBD) [51]. Disruption of the gut barrier increases the influx of immunostimulatory microbial ligands into the systemic circulation, potentially leading to obesity, diabetes [52] and MASLD [53]. Gut bacteria that produce components from either dietary or host-derived glycans that enhance gut barrier function and thereby reduce intestinal permeability may serve as a novel therapeutic target for systemic inflammation in metabolic diseases [54]. For example, *Akkermansia muciliphila*, which is abundant in the mucosal lining, functions as a guardian of the gut barrier but also has protective effects against obesity and diabetes [55].

A causal relationship between the gut microbiota and host metabolism was first demonstrated in FMT studies. The gut microbiota composition at baseline was shown to mediate the improvement in insulin sensitivity in FMT-treated metabolic syndrome [56]. This study suggests the microbiome as a mediator of insulin sensitivity regulation. Similar results were obtained when fecal material from lean donors was transferred to subjects with metabolic syndrome, which was associated with increased activities of butyrate-producing bacteria in the small intestine [15]. Gut microbiota and their metabolites have been suggested to modulate host epigenetic alterations in MASLD [57]. A randomized clinical trial showed that FMT treatment delayed the progression of MASLD by altering the gut microbiome, and the beneficial effects were more pronounced in lean MASLD than in obese MASLD individuals [58]. Due to the risk of introducing unwanted components from the fecal samples during FMT treatment [59], efforts have been made to construct synthetic bacterial communities that resemble the healthy microbiome [60]. A mixture of 17 *Clostridium* species induced colonic regulatory T cells in mice [61]. Some success has been achieved in the treatment of *Clostridioides difficile* infection [62,63]. This approach is promising but also presents challenges due to the complexity of the microbial communities that need to be maintained. The fermentation and production of multiple strains for preclinical and clinical studies are other issues. Some encouraging data have been obtained from single bacterial strain treatments to improve metabolic performance in mice and humans (Figure 3). These are discussed in the following sections.

## 5. Potential Therapeutic Bacteria for Metabolic Health

*Akkermansia muciniphila* is a mucin-degrading bacterium that resides abundantly in the mucus layer of mammals [64]. Metagenomic analyses revealed a high phylogenetic and functional diversity of *Akkermansia* in humans [65]. This bacterium is capable of degrading mucin and sugars to produce the short chain fatty acid propionate [66], which can bind to G-protein receptors expressed by L-cells, thereby stimulating GLP-1 secretion. Its unique niche on the mucus layer allows this bacterium to interact directly with the host. *A. muciniphila* has been shown to improve epithelial barrier function via secreted components and outer membrane components, contributing to health-promoting effects [67,68]. Lower levels of *A. muciniphila* have been associated with obesity and diabetes [40,69]. Supplementation with live *A. muciniphila* reversed high-fat-diet-induced metabolic disorders, including body weight gain, adipose tissue inflammation and insulin resistance in mice [69]. Interestingly, anti-obesity and anti-diabetes effects have also been observed with pasteurized *A. muciniphila* and an outer membrane protein Amuc-1100 [67], opening up new therapeutic options to target obesity and diabetes. A proof-of-concept study in obese and overweight individuals showed that daily bacterial administration for 3 months improved insulin sensitivity and reduced insulinemia and plasma cholesterol with a small reduction in body weight [13]. Furthermore, metformin has been reported to increase the relative abundance of *A. muciniphila* in vitro and in humans [44,70], supporting a protective role of this bacterium in metabolic disorders. Taken together, *A. muciniphila* has a great potential as a therapeutic bacterium for metabolic health.

Short chain fatty acid butyrate, as discussed above, is clearly an important signaling molecule with proven protective effects against several diseases [21,71]. A reduced abundance of butyrate-producing bacteria has been repeatedly observed in type 2 diabetes [72,73,74]. Therefore, it is plausible that butyrate-producing bacteria (either single strains or mixtures) are viable targets for the therapeutic treatment and prevention of metabolic diseases. To date, several butyrate-producing species have shown effects on improving insulin sensitivity in mice and humans. *Anaerobutyricum soehngenii* is the most advanced in the development and is currently in Phase II clinical trials. It was first observed that the level of *E. hallii* (formal nomenclature of *A. soehngenii*) in the small intestine was associated with improved insulin sensitivity after an FMT from lean subjects to those with metabolic syndrome [15]. Similar effects were subsequently observed in an efficacy study in mice [14] and in a mechanistic study in humans [75]. In this mechanistic study, the authors showed that duodenal perfusion of *A. soehngenii* in subjects with metabolic syndrome increased the expression and secretion of GLP1, which was associated with increased fecal butyrate levels. A recent human study showed that the administration of *A. soehngenii* capsules for release in the small intestine improved glycemic controls in metabolic syndrome [43]. Taken together, these data suggest that *A. soehngenii* is a potential therapeutic microbe for metabolic disorders.

*Anaerostipes* is another abundant butyrogenic species that has been associated with reduced metabolic risk in large human cohorts [37,76]. The administration of *Anaerostipes rhamnosivorans* for 6 weeks reduced fasting glucose levels and did not affect insulin sensitivity in diet-induced obese mice [76]. A newly identified butyrate-producing species, *Dysosmobacter welbionis*, was shown to reduce body weight gain and improve glucose metabolism only after at least 9 weeks of supplementation [77]. It is therefore of interest to investigate whether insulin sensitivity and other metabolic parameters can be improved by prolonged oral supplementation with *A. rhamnosivorans*. It has recently been discovered that *Anaerostipes* forms a synergistic interaction with a prevalent gut microbe, *Mitsuokella jalaludinii*, in the conversion of dietary phytate, an abundant bioactive component in plants, to propionate [78]. As plant-based diets and propionate have been associated with metabolic benefits, further studies are awaited to investigate the therapeutic potential of combining these two bacteria with phytate for metabolic health.

There are several species that have been shown to have anti-obesity and anti-diabetic effects. *Christensenella minuta* has been associated with a lean phenotype in a British twin cohort [79]. In the same study, oral supplementation with this bacterium reduced body weight gain and fat mass once administered in obese humanized mice as compared to placebo (obese humanized mice without bacterial supplementation). Reduced levels of *C. minuta* have been observed in subjects with metabolic disorders [80,81], and the levels of this bacterium have been inversely associated with several metabolic risk biomarkers, including BMI, triglycerides and high-density lipoprotein [82]. The observed effects of this bacterium are mostly from mouse studies and no human data have been reported yet. *Intestinimonas butyriciproducens* is another butyrogenic species with a unique metabolism that has the potential to be used as a therapeutic strain for metabolic diseases. It is by far the only gut bacterium that is capable of converting dietary fructoselysine to butyrate [83]. Fructoselysine is an abundant Amadori product formed in cooked foods and is also an intermediate in the formation of advanced glycation end products (AGEs), harmful compounds associated with high cardiometabolic risk [84]. The conversion of fructoselysine to butyrate by *I. butyriciproducens* is highly favorable given the importance of butyrate for metabolic health. This is supported by the observation that the abundance of fructoselysine pathway genes was inversely associated with BMI, fasting insulin and triglycerides in a human cohort, and that 13 weeks of *I. butyriciproducens* administration reduced body weight gain and fat mass and improved insulin sensitivity in a diet-induced obesity mouse model [85]. This was consistent with reduced levels of plasma butyrate in treated mice as compared to placebo. *Parabacteroides distasonis* has been shown to alleviate obesity and metabolic dysfunction in mice via the production of succinate and secondary bile acids [86]. The oral administration of *Odoribacter laneus* was found to improve glucose control and inflammatory profile in obese mice by depleting circulating succinate [87]. Furthermore, supplementation with *Bacteroides thetaiotaomicron* protected mice against obesity in a high-fat-diet-induced obesity model [36]. Engineered bacterial strains have been reported to reprogram the intestinal cells to improve glucose response and insulin secretion [88]. This is an interesting approach with great potential, but the regulation of the use of genetically modified bacteria is strict and requires careful consideration of the use of these strains as therapeutic microbes.

## 6. Microbial Components Affect Host Metabolism

Crosstalk between the human microbiota and the host occurs either by direct contact or indirectly through the secretion of components that allow communication between the bacteria and host cells. A number of microbial metabolites have been identified that are associated with the risk of developing diabetes have been identified [89] and many others are associated with obesity, MASLD and T2D [90]. Many newly identified metabolites or components remain to be characterized while the well-studied microbial components in Figure 4 are discussed below.

*Bile acids* are important signaling molecules that have been shown to crosstalk with the human microbiota, thereby affecting host metabolism [91]. Primary bile acids are produced in the liver and released into the duodenum during fat digestion, and the microbiome converts the primary bile acids into secondary bile acids, changes in which have been linked to inflammatory bowel disease [92] and intestinal carcinogenesis [93]. Bile acids have been shown to activate the Takeda G protein-coupled receptor 5 (TGR5), a G-protein-coupled receptor and the farnesoid X receptor FXR, a nuclear hormone receptor; thereby inducing GLP1 release, leading to improved glucose homeostasis [94,95]. Recently, new conjugated bile acids derived from the microbiome have been reported, and these new bile acids have shown activity at the FXR receptor in vitro [96]. A new microbial bile salt hydrolase responsible for acyl transformation has recently been discovered [97], further contributing to the expansion of bile acid diversity. An increased abundance of novel bile acid pathways has been observed in the microbiome of centenarians [98], suggesting that this bile acid synthesis may be associated with longevity. However, the role of these newly identified bile acids or enzymes in health and disease remains to be elucidated.

*Tryptophan* is an essential amino acid that can only be obtained from food. Gut bacteria have been shown to play a central role in tryptophan metabolism, with the ability to convert tryptophan to various molecules, such as indole and its derivatives, thereby contributing to maintain intestinal homeostasis [99]. For example, *Clostridium sporogenes* has been reported to convert tryptophan to both tryptamine and indole-3-propionic acid [100], whereas *Ruminococcus gnavus* only produces tryptamine from the decarboxylation of tryptophan [101]. Indole propionic acid levels have been reported to be associated with a lower risk of developing T2D [102]. In addition, indole-3-propionic acid has been found to regulate intestinal barrier function by interacting with the pregnane X receptor [103] and to improve glucose metabolism in rats [104]. Previous studies have shown that bacterial indole promotes the release of glucagon-like peptide-1 (GLP1) to slow down gastric emptying and reduce the appetite [105]. Indoles produced by the gut microbiota have also been reported to activate the aryl hydrocarbon receptor (AhR) [106], a transcription factor involved in immune regulation and cytokine release, thereby contributing to gut health. As a product of microbial tryptophan metabolism, gut microbiota-derived tryptamine has been shown to impair insulin signaling in animal models by activating the activating MAPK/ERK pathway [107]. A recent study showed that FMT halted the progression of onset type 1 diabetes in which 6-bromo-tryptophan, a metabolite derived from microbial tryptophan metabolism, was associated with prolonged residual ß-cell function [108]. These data suggest that the microbial production of these metabolites may be beneficial for metabolic health. Therapeutic approaches to deliver beneficial tryptophan-derived metabolites have great potential and require validation in mice and humans.

*Short chain fatty acids* (SCFAs) are end metabolites of bacterial fermentation of indigestible dietary components, mainly fiber [21]. They include butyrate, propionate and acetate, which are mainly produced in the caecum and colon in a molar ratio of 1:1:3 [109,110]. SCFAs act as signaling molecules and enter the systemic circulation, thereby affecting the metabolism and function of various peripheral tissues [111]. SCFAs have been shown to activate GPCR receptors (GPR41/43), thereby inducing the secretion of the incretin hormones PYY and GLP1 [112,113], which may contribute to improved glucose metabolism and insulin secretion [114]. SCFAs produced from the fermentation of dietary fiber have been found to activate intestinal gluconeogenesis via a gut–brain neural circuit, providing metabolic benefits for body weight and glucose control [115]. In addition, epithelial barrier function and intestinal permeability were also improved by SCFA exposure through modulation of tight junction protein expression [116,117,118]. Previous studies have shown that maintaining epithelial integrity with low permeability is crucial to prevent leakage of toxic components into the systemic circulation, which can cause chronic inflammation, weight gain and insulin resistance [119,120]. The SCFA butyrate has been shown to interact directly with host receptors to suppress colonic inflammation and carcinogenesis [71]. With all the metabolic benefits observed above, many studies have been conducted to investigate the potential benefits of SCFA supplementation on metabolic health. It has been reported that the oral supplementation of SCFA for 4 weeks improved hepatic metabolic functions via the FFAR3 receptor in mice [121], while 12 weeks of SCFA administration prevented diet-induced obesity in mice through the regulation of GPCR receptors in adipose tissue and colon [122]. The intraperitoneal administration of SCFAs improved lipid metabolism in rats [123]. In addition, mice fed a high-fat diet and treated with tributyrin, a butyrate precursor drug, were protected from diet-induced obesity, insulin resistance and hepatic steatosis [124]. Oral supplementation with butyrate was found to reduce fasting insulin levels in diet-induced obese mice [125]. This is in contrast to a human study in which butyrate supplementation had no beneficial effect on glucose metabolism in subjects with metabolic syndrome [126]. Propionate administration for 22 weeks reduced hepatic lipogenesis and improved insulin sensitivity in a diet-induced obesity mouse model [123]. Two oral supplements of propionate increased resting energy expenditure and lipid oxidation in fasted humans [127]. The discrepancy between animal and human studies warrants the need for well-designed studies to further investigate the benefits of SCFAs in humans. Furthermore, the administration of these short chain fatty acids has produced inconsistent results, which may be due to their rapid absorption in the upper gastrointestinal tract, whereas all SCFA receptors are predominantly located in the lower gastrointestinal tract and colon [128]. While supplementation with SCFAs does not solve the problem of an imbalanced microbiome and must be carried out on a regular basis, the administration of bacterial strains would be more sustainable in this context.

There is an increasing number of microbial metabolites that signal to the host, and *Akkermansia muciniphila* has been shown to improve barrier function and metabolic health in a variety of mouse models [55].The main mode of action has been attributed to its outer membrane protein Amuc_1100, which can bind to Toll-like receptor-2 (TLR-2) to stimulate expression of tight junction proteins and improve barrier function in mice [67]. As this Amuc_1100 protein is thermostable, this rationalizes the similar effect of pasteurized *A. muciniphila* administration in mice. It is noteworthy that a recent human study showed improved barrier function following the administration of pasteurized *A. muciniphila* in obese humans with metabolic syndrome [13]. In addition, a newly identified protein P9 secreted by *A. muciniphila* can bind to ICAM-2 to directly induce the L-cell secretion of GLP1 in mice [68]. The lack of increased GLP1 activity in the human studies suggests that the efficacy in humans remains to be established. Microbial imidazole propionate derived from histidine impaired glucose tolerance and insulin signaling via the activation of mTORC1 [129] and negatively affected metformin action via the inhibition of AMPK activity [130] in mice. Microbiome-derived ethanol has been shown to contribute to the pathogenesis of non-alcoholic fatty liver disease [28]. The oral supplementation of an alcohol-producing strain has been shown to induce fatty liver disease in mice [131]. New approaches to block microbial ethanol production in the gut may offer therapeutic potential in the prevention and treatment of liver disease. In addition, succinate produced by gut bacteria activated intestinal gluconeogenesis, thereby improving glycemic control in mice [132]. Future studies in humans are needed to verify the effects of these newly identified microbial components.

## 7. Conclusions and Future Perspectives

With the rapid increase in the incidence and prevalence of metabolic disorders such as obesity, type 2 diabetes and non-alcoholic fatty liver disease, there is an urgent need for effective therapeutic options to prevent disease progression. It is clear that the human microbiome is strongly associated with metabolic health. There is increasing evidence of causal links between the human microbiome and metabolic disorders in humans and rodents, which holds great promise for the potential use of microbiome-based therapeutics to intervene in host metabolic performance.

Recent advances in sequencing technologies allow in-depth studies of the human microbiome with pathologies of multiple diseases, generating hypotheses for potential bacterial candidates as well as microbial components for disease prevention and treatment. Yet, future research is needed to validate all these hypotheses in well-designed mechanistic and efficacy studies in animal, human and in vitro cell models. The choice of whether to use live or dead bacteria, and the dose of the treatment, will also need to be considered for clinical interventions to achieve optimal benefit. It is certainly necessary to establish more complete reference databases for the identification of novel metabolites or peptides, enabling the discovery of new microbiome-derived components that link the microbiome and host metabolism. Most current human studies rely on the analysis of the fecal microbiome, which does not represent the upper gut microbiome, an important regulator of human health [133]. Therefore, smart sampling capsules with sensing technologies with minimally invasive approaches that allow the measurement of pH, temperature and pressure and collect intestinal samples would be greatly appreciated. It has become increasingly clear that most drugs affect the microbiome and metabolome profiles in addition to their known clinical effects. And perhaps the combination of different bacterial strains with dietary components or drugs can have synergistic effects on host metabolism.

## Figures and Tables

**Figure 1 nutrients-16-02322-f001:**
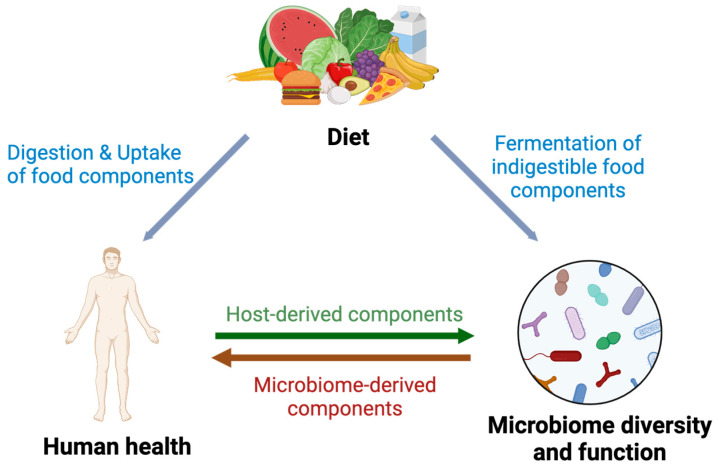
Complex interplay between the gut microbiome, diet and host health. Diet provides nutrients that are absorbed directly by the host and metabolized by the microbiome via fermentation, resulting in the production of metabolites that affect the host. In turn, the host provides its own components to the microbiome, such as mucin and mucus-derived glycans.

**Figure 2 nutrients-16-02322-f002:**
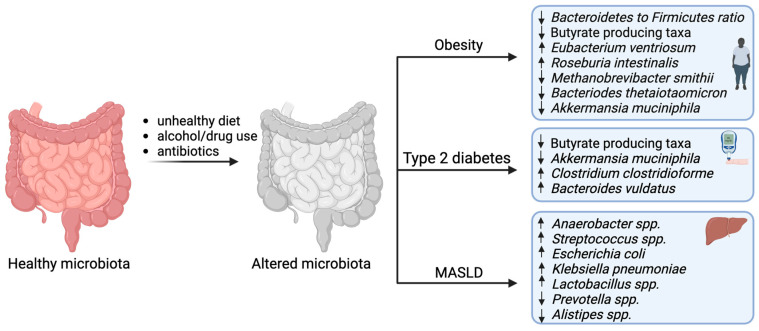
Altered gut microbiota in metabolic disease. Unhealthy diet, alcohol or drug use and antibiotics alter the composition and function of the microbiota in unfavorable ways, which may contribute to the development of metabolic disorders. Despite the wide variation in the pathologies of chronic metabolic disorders, a few microbial groups are commonly reduced in individuals with metabolic disorders such as butyrate-producing taxa and *Akkermansia muciniphila*. These bacterial species may therefore be potential therapeutic targets for the treatment and prevention of metabolic disorders. Down arrow as reduction and up arrow as increase.

**Figure 3 nutrients-16-02322-f003:**
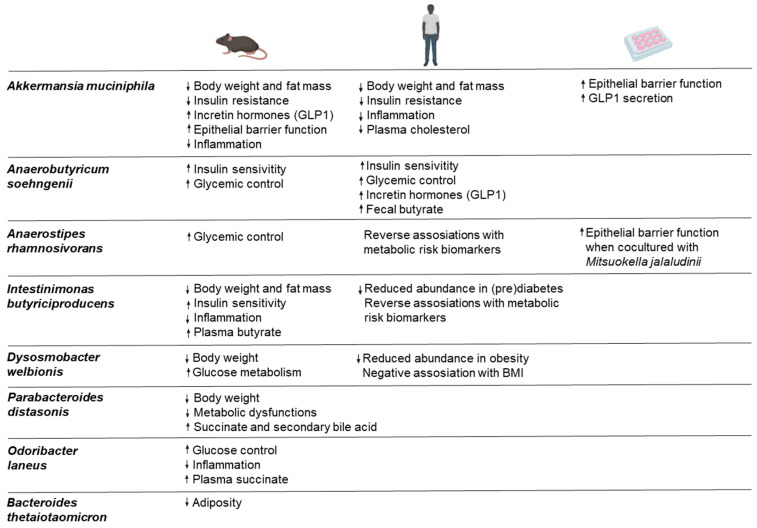
Potential therapeutic gut microbes with proven metabolic health benefits. GLP1: Glucagon-like peptide 1; BMI: Body mass index. Down arrow as reduction and up arrow as increase.

**Figure 4 nutrients-16-02322-f004:**
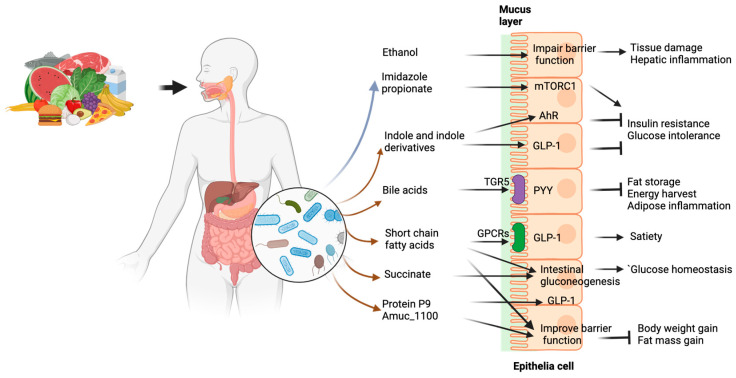
Microbial components regulate host metabolism. Components with a positive influence on host metabolism are indicated by brown arrows, while components with a negative influence are indicated by grey arrows.

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
