# Peer review of "The Human Microbiome as a Therapeutic Target for Metabolic Diseases"

_nutrients, 2024, doi:10.3390/nu16142322_

Round 1

Reviewer 1 Report

Comments and Suggestions for Authors

In this work, author introduced the human microbiome as therapeutic target for metabolic disease. This is an interesting topic. There are some questions need to solved before publication:

1. In the section 3 "Altered Microbiota in Metabolic Diseases ", a illustrate or figure should be drawn and to present the relationship between gut microbiota and metabolic diseases.

2. Among these metabolic disorders (including T2D, obesity, MASLD, and so on), is there a commonality or trend in the changes of gut microbiota or the specific bacteria, which could provide valuable insights for targeted therapies in subsequent treatments?

3. In the section 4 "The Human Microbiome as Therapeutic Target to Improve Metabolic Health ", I missed the introduction of MASLD, which is also closely related with gut microbiota and gut barrier.

4. In the line 227-234 on section 5, Faecalibacterium praunitzii  were mentioned for IBD treatment, however, IBD may not belong to metabolic diseases, please correct.

5. Generally, "via" should be presented using italic, please go through manuscript and revise it.

6. In line 325, the abbreviation of Short chain fatty acids  should be SCFAs generally, please correct it in manuscript.

7. The limitations of using microbiome as therapeutic target for metabolic diseases should be presented in section 7, such as using live or die bacteria to treatment, or how choose the treatment doses.

Comments on the Quality of English Language

Moderate editing of English language required

Author Response

In this work, author introduced the human microbiome as therapeutic target for metabolic disease. This is an interesting topic. There are some questions need to solved before publication:

Comment 1: In the section 3 "Altered Microbiota in Metabolic Diseases ", a illustrate or figure should be drawn and to present the relationship between gut microbiota and metabolic diseases.

Response 1: The author has included a figure illustrating the difference of the microbiome between healthy versus metabolic diseases including obesity, T2D and MASLD in the revised manuscript.

Comment 2: Among these metabolic disorders (including T2D, obesity, MASLD, and so on), is there a commonality or trend in the changes of gut microbiota or the specific bacteria, which could provide valuable insights for targeted therapies in subsequent treatments?

Response 2:  Despite the large discrepancy of the microbiome composition between individuals, it has become clear that the abundance of butyrate producing species and Akkermansia muciniphila was reduced in obese and T2D subjects discussed in the text and the additional figure while clear associations between these species with MASLD have not been reported yet. As obesity is a risk factor for MASLD, it is plausible that supplementation of butyrate/producing species and A. muciniphila may be beneficial in MASLD. This discussion has been included in the legend of the added figure as Figure 2 in the revised manuscript. Please see highlighted lines156-158 and  181-188.

Comment 3: In the section 4 "The Human Microbiome as Therapeutic Target to Improve Metabolic Health ", I missed the introduction of MASLD, which is also closely related with gut microbiota and gut barrier.

Response 3: The author has added the introduction of MASLD with a recent FMT clinical trial for MASLD and discussed the gut homeostasis in MASLD in the revised manuscript. Please see highlighted lines in 228, 235, 248-253.

Comment 4: In the line 227-234 on section 5, Faecalibacterium praunitzii  were mentioned for IBD treatment, however, IBD may not belong to metabolic diseases, please correct.

Response 4: It is correct that IBD is not considered as a type of metabolic diseases but subjects with metabolic diseases are often observed to have intestinal and systemic inflammation caused by disrupted intestinal epithelial barrier. Thereby, bacteria that are beneficial in IBD and can be used to reduce inflammation may be beneficial in metabolic diseases. Nevertheless, it may cause some confusion, hence the author agrees to leave out F. praunitzii from the text (line 322) and also the table in the revised manuscript.

Comment 5: Generally, "via" should be presented using italic, please go through manuscript and revise it.

Response 5: The author thanks the reviewer for the correction. This has been corrected thorough the revised manuscript.

Comment 6: In line 325, the abbreviation of Short chain fatty acids  should be SCFAs generally, please correct it in manuscript.

Response 6: The author thanks the reviewer for the correction. This has been corrected in the revised manuscript. See highlighted text from 429-469.

Comment 7: The limitations of using microbiome as therapeutic target for metabolic diseases should be presented in section 7, such as using live or die bacteria to treatment, or how choose the treatment doses.

Response 7: The author thanks the reviewer for the correction. The author has added this in the “conclusion and future perspectives” in the revised version of the manuscript. See lines 510-512.

Reviewer 2 Report

Comments and Suggestions for Authors

The reviewer acknowledges the authors of the manuscript entitled "The human microbiome as therapeutic target for metabolic diseases". This is a review on a relevant topic nowadays and covering a good range of the aspects to be reviewed under that thematic. on an overall revision the reviewer as to state that some spelling mistakes can be found on the manuscript, as well as phrases constructing or incorrect English usage. the reviewer start to point the grammar and writing alterations to be made:

line 36: should be "overuse of processed food" instead of "overuse of process food"

line 41 to line 47: references should be added to support that information

line 174 to line 175: " transfers of undesired components from fecal material" is repeated

table 1: definition of GLP1 and BMI should be added similar to what was done for IBD

line 227 and 228: something is missing to link the 2 ideas written on this sentence

line 245: should be "plant-based diets"

line 384 to 386: sentence is not correctly written, maybe authors would like to say: "due to the rapid rise in incidence and prevalence of metabolic disorders including obesity, type 2 diabetes and non-alcoholic fatty liver disease, there is an urgent need of effective therapeutic options for preventing disease progression"

Additionally the reviewer would like to see some more recent literature to be used on this review like the clinical trial described in the manuscript with the doi: 10.7717/peerj.17583, the review with the doi:10.1136/gutjnl-2024-332398 and some literature described on it or the manuscript with the doi: 10.3390/antiox13060746.

Finally, this reviewer would like again to acknowledge the authors of this manuscript for the effort on revising this thematic.

Comments on the Quality of English Language

Some minor editing should be made on the English usage and on the sentences construction according to the suggestions that the reviewer left for the authors.

Author Response

The reviewer acknowledges the authors of the manuscript entitled "The human microbiome as therapeutic target for metabolic diseases". This is a review on a relevant topic nowadays and covering a good range of the aspects to be reviewed under that thematic. on an overall revision the reviewer as to state that some spelling mistakes can be found on the manuscript, as well as phrases constructing or incorrect English usage. the reviewer start to point the grammar and writing alterations to be made:

Comment 1: line 36: should be "overuse of processed food" instead of "overuse of process food"

Response 1: This has been corrected in the revised version of the manuscript. See line 77.

Comment 2: line 41 to line 47: references should be added to support that information

Response 2: The author assumes the reviewer meant 81-87 for the missing references. The author has added references to these statements in the revised version of the manuscript. Please see line 81 and 84.

Comment 3: line 174 to line 175: " transfers of undesired components from fecal material" is repeated

Response 3: The author apologises for this duplication. The duplicated text has been removed from the revised manuscript. See lines 253-255.

Comment 4: table 1: definition of GLP1 and BMI should be added similar to what was done for IBD

Response 4: The author has added the definition of GLP1 and BMI to the table legend.

Comment 5: line 227 and 228: something is missing to link the 2 ideas written on this sentence

Response 5: A sentence has been added at the beginning of the paragraph and the following sentence has been rephrased. Please see lines 242-245

Comment 6: line 245: should be "plant-based diets"

Response 6: The author assumes the reviewer meant the mis-spelling in line 316. This has been corrected in the revised manuscript. See line 335.

Comment 7: line 384 to 386: sentence is not correctly written, maybe authors would like to say: "due to the rapid rise in incidence and prevalence of metabolic disorders including obesity, type 2 diabetes and non-alcoholic fatty liver disease, there is an urgent need of effective therapeutic options for preventing disease progression"

Response 7: The author thanks the reviewer for pointing out this typo-mistake. The sentence has been corrected in the revised manuscript. See lines 497-499.

Comment 8: Additionally the reviewer would like to see some more recent literature to be used on this review like the clinical trial described in the manuscript with the doi: 10.7717/peerj.17583, the review with the doi:10.1136/gutjnl-2024-332398 and some literature described on it or the manuscript with the doi: 10.3390/antiox13060746.

Response 8: The author thanks the reviewer for the suggestion. Although 3 suggested papers are of interest, the study design of the clinical trial is still questionable and the link between hydrogen-rich coral calcium and the modification of the microbiome to induce MASLD was far-fetched. Therefore, the author only added the review on MASLD and another FMT study in MASLD in the revised manuscript in lines 248-252. The author hopes the reviewer would understand.

Comment 9: Finally, this reviewer would like again to acknowledge the authors of this manuscript for the effort on revising this thematic.

Response 9: Thank you for your acknowledgement.

Reviewer 3 Report

Comments and Suggestions for Authors

I congratulate the author for the work

The topic is very interesting and current

The paper is well written and only some minor corrections are due

Figures are nice

These are my comments that I hope can improve the paper

line 29-30 missing reference

line 71-72 the sentence about increase in transit time is not correct

line 94 please correct "refine" with "refined"

line 154-157 please add a reference

lne 261-262 please rephrase "toxic compounds with high cardiometabolic risks "

line 305 in this paragraph about tryptophan, please add some examples of bacteria able to convert it to I3P and tryptamine

line 366 please correct the double space

line 384-386 please reformulate the sentence

Comments on the Quality of English Language

English language is fine and only some corrections are required

Author Response

I congratulate the author for the work

The topic is very interesting and current

The paper is well written and only some minor corrections are due

Figures are nice

These are my comments that I hope can improve the paper

Comment 1: line 29-30 missing reference

Response 1: The ref has been added. See line 76

Comment 2: line 71-72 the sentence about increase in transit time is not correct

Response 2: The sentence has been modified. See highlighted lines 112-114.

Comment 3: line 94 please correct "refine" with "refined"

Response3: This has been corrected in the revised manuscript. See line 142.

Comment 4: line 154-157 please add a reference

Response 4: The author assumes the reviewer asked for the references of the statement written in lines 199-201 which has been included in the revised manuscript.

Comment 5: line 261-262 please rephrase "toxic compounds with high cardiometabolic risks "

Response 5: It has been rephrased as “harmful compounds associated with high cardiometabolic risks”. See highlighted text in lines 353-355.

Comment 6: line 305 in this paragraph about tryptophan, please add some examples of bacteria able to convert it to I3P and tryptamine

Response 6: The author thanks the reviewer for the suggestion. Examples of bacteria for tryptophan conversion to I3P and tryptamine have been added. Please see highlighted text in lines 408-411.

Comment 7: line 366 please correct the double space

Response 7: corrected

Comment 8: line 384-386 please reformulate the sentence

Response 8: The sentence has been rephrased in highlighted text (lines 487-488) in the revised manuscript.

Reviewer 4 Report

Comments and Suggestions for Authors

The review is informative and well-summarised with the latest information on the subject matter.

Just one suggestion, please provide the details of the service used for image preparation as acknowledgement.

Author Response

The review is informative and well-summarised with the latest information on the subject matter.

Comment 1 Just one suggestion, please provide the details of the service used for image preparation as acknowledgement.

Response 1 The author has added Biorender.com in the acknowledgment.